# Structure-Function Analyses of New SARS-CoV-2 Variants B.1.1.7, B.1.351 and B.1.1.28.1: Clinical, Diagnostic, Therapeutic and Public Health Implications

**DOI:** 10.3390/v13030439

**Published:** 2021-03-09

**Authors:** Jasdeep Singh, Jasmine Samal, Vipul Kumar, Jyoti Sharma, Usha Agrawal, Nasreen Z. Ehtesham, Durai Sundar, Syed Asad Rahman, Subhash Hira, Seyed E. Hasnain

**Affiliations:** 1JH-Institute of Molecular Medicine, Jamia Hamdard, New Delhi 110062, India; jasdeep002@gmail.com; 2ICMR National Institute of Pathology, Safdarjung Hospital Campus, New Delhi 110029, India; samaljasmine@gmail.com (J.S.); jyoti22khandal@gmail.com (J.S.); uburra@gmail.com (U.A.); nzehtesham@gmail.com (N.Z.E.); 3Department of Biochemical Engineering and Biotechnology, Indian Institute of Technology, New Delhi 110016, India; vipul.kumar@dbeb.iitd.ac.in; 4BioInception Pvt. Ltd., Swift House Ground Floor, 18 Hoffmanns Way, Chelmsford, Essex CM1 1GU, UK; 5Department of Global Health, University of Washington-Seattle, Seattle, WA 98195, USA; 6Dr Reddy’s Institute of Life Sciences, University of Hyderabad Campus, Prof. C.R. Rao Road, Gachibowli, Hyderabad 500049, India

**Keywords:** B.1.1.7, B.1.351, B.1.1.28.1, 501Y.V1, 501Y.V2, P.1, Clade G, COVID-19 vaccines, D614G variant, furin cleavage site, immune escape, ORF8, spike protein, public health strategies, vaccine delivery

## Abstract

SARS-CoV-2 (Severe Acute Respiratory Syndrome-Coronavirus 2) has accumulated multiple mutations during its global circulation. Recently, three SARS-CoV-2 lineages, B.1.1.7 (501Y.V1), B.1.351 (501Y.V2) and B.1.1.28.1 (P.1), have emerged in the United Kingdom, South Africa and Brazil, respectively. Here, we have presented global viewpoint on implications of emerging SARS-CoV-2 variants based on structural–function impact of crucial mutations occurring in its spike (S), ORF8 and nucleocapsid (N) proteins. While the N501Y mutation was observed in all three lineages, the 501Y.V1 and P.1 accumulated a different set of mutations in the S protein. The missense mutational effects were predicted through a COVID-19 dedicated resource followed by atomistic molecular dynamics simulations. Current findings indicate that some mutations in the S protein might lead to higher affinity with host receptors and resistance against antibodies, but not all are due to different antibody binding (epitope) regions. Mutations may, however, result in diagnostic tests failures and possible interference with binding of newly identified anti-viral candidates against SARS-CoV-2, likely necessitating roll out of recurring “flu-like shots” annually for tackling COVID-19. The functional relevance of these mutations has been described in terms of modulation of host tropism, antibody resistance, diagnostic sensitivity and therapeutic candidates. Besides global economic losses, post-vaccine reinfections with emerging variants can have significant clinical, therapeutic and public health impacts.

## 1. Background

Since the initial outbreak of COVID-19, the Severe Acute Respiratory Syndrome-Coronavirus 2 (SARS-CoV-2) virus has claimed more than 2.4 million lives out of 100 million affected individuals. The SARS-CoV-2 genome codes for non-structural (nsp) and structural proteins including the spike (S), nucleocapsid (N), membrane (M), and envelope (E) proteins [1,2]. The S protein mediates initial contact with human hosts while the E and M proteins function in viral assembly and budding. The recent temporal analyses of SARS-CoV-2 epidemics highlighted selective global sweep of the D614G variant S protein (Clade G) over G251V in ORF3a (Clade V) and L84S in ORF 8 (Clade S) variants [2,3,4,5]. The ubiquitous D614G variant of SARS-CoV-2 exhibits efficient replication in upper respiratory tract epithelial cells and higher transmissibility among humans, thereby conferring enhanced fitness [6,7]. As per the latest global reports on COVID-19, three new strains assigned to lineages 501Y.V1, 501Y.V2 and P.1 have been identified (Figure 1A–C) (cov-lineages.org). The former, referred to as SARS-CoV-2 VOC 202012/01 (Variant Of Concern, year 2020, month 12, variant 01), was identified as a part of virological and epidemiological analysis, due to a sudden rise in COVID-19 cases detected in south-east England (Figure 1A) [8,9]. For week 51 of 2020, hospital and/or intensive care unit (ICU) occupancy and/or new admissions due to COVID-19 were high (at least 25% of the peak level during the pandemic) or had increased compared with the previous week in the UK and other 30 countries [10]. For week 51 of 2020, hospital and/or ICU occupancy as well as new admissions due to COVID-19 had increased compared with the previous week in the UK and other 30 countries [10]. For week 52/2020, all-cause excess mortality data from the UK and EU/EEA countries reported to the EuroMOMO network identified a recent substantial increase in mortality, mainly affecting those aged 45 years and above and likely attributed to the 501Y.V1 variant. Preliminary reports from the UK suggested higher transmissibility (increase by 40–70%) of this strain, escalating the R_o_ (basic reproduction number) of the virus to 1.5–1.7 [8,11]. This apparent fast spreading variant shows twenty three mutations—thirteen non-synonymous, six synonymous and four amino acid deletions and is reported by forty-five nations [8]. The 501Y.V2 lineage emerged in the Nelson Mandela Bay area of Eastern Cape Province, South Africa, followed by its steep spread to Eastern and Western Cape Provinces (Figure 1B) [12]. In mid-October after gradual weakening on first epidemic wave, the Nelson Mandela Bay area showed 20% PCR positivity rate followed by resurgence of a second wave in both Eastern and Western cape provinces, resulting in R_o_ > 1 [12]. The identified mutant strain (501Y.V2) displays nine non-synonymous mutations along with three amino acid deletions and is reported by twenty-four countries till date. Another variant from Brazil (known as VOC202101/02 in UK), identified first in Japanese travelers from Brazil, shows seventeen unique mutations including the N501Y and E484K mutations [13]. As of 23 January 2021, the P.1 lineage has been reported by six countries, including Germany, Italy, Brazil, Japan and South Korea (Figure 1C) [14,15]. 

In the current scenario, where immunization programs have already commenced in nations highly affected by COVID-19, the advent of new variants has raised global public health concerns worldwide on the possible role in disease severity and antibody responses. An important question that raises the alarm is what if these new variants are “immune escape” variants, which means, people who have had SARS-CoV-2 infection are susceptible to get re-infection and therefore, the current vaccines probably need redesigning to be effective against the variants. The current report highlights crucial non-synonymous mutations and deletions occurring in SARS-CoV-2 501Y.V1, 501Y.V2 and P.1 variants and their potential impact on the overall structure–function of SARS-CoV-2 proteins. Certain mutations/deletions have been previously associated with increased antibody resistance (decrease in binding affinity) to target protein(s) (Table 1) which could affect polyclonal antibody response elicited by infections/ vaccinations. As of 20 January 2021, ten vaccine candidates are in clinical use globally. Global trial maps of major vaccine candidates show co-emergence of these mutant variants (Figure 1A–C). The major vaccine candidates are likely to confer immunity, although reduced protection might be a concern [13]. Dissection of functional impact of these mutations highlights their immediate relevance to global health in terms of viral diagnosis, clinical and public health strategies. 

## 2. Methods

### 2.1. Structural Analysis and Protein-Protein Interactions

Structural effects of SARS-CoV-2 mutations were studied using variant analysis module of COVID-3D suite [16,18]. Images were created using PyMol [19]. The structure of the N protein of SARS-COV-2 was obtained through Swiss-model interactive modelling (swissmodel.expasy.org/ accessed on December 30, 2020). The high-quality model was generated using 1.45 Å resolution crystal structure of its C-terminal dimerization domain (PDB id: 6yun) and NMR solution structure of its N-terminal domain in complex with 7mer dsRNA (PDB id: 7acs). Protein-protein docking of the S protein with transmembrane protease serine 2 (TMPRSS2) was carried out using Patchdock rigid docking using clustering RMSD of 0.4 [20]. The solutions were refined using FireDock which addresses protein flexibility and scoring of solutions obtained through Patchdock [21,22]. Protein–protein complexes of S protein receptor binding domain (RBD) (both wildtype and mutants) with anti-SARS-CoV-2 neutralizing CR3022 (IgM type) C135 (IgG type) antibodies were prepared through swiss-model interactive modelling. The resulting structures were reanalyzed for obtaining global docking scores between RBD and antibody complexes using FireDock refinement. 

### 2.2. Molecular Dynamics Simulations Set Up and Runs

The Molecular Dynamics (MD) simulation of the wildtype and mutant receptors as well as receptor antibody complexes were done using the Desmond MD tool integrated with Maestro Schrodinger software [23]. The systems were first prepared using the System builder tool in Desmond Schrodinger suite. For this, the TIP3P solvation model was used and the system boundary was generated using a 10 Å^3^ box. The systems were neutralized by adding appropriate numbers of ions using OPLS3e force field [24]. Each system was solvated with the TIP3P water model in an orthorhombic box with periodic boundary conditions. To prevent interaction of the protein complex with its own periodic image, the distance between the complex and the box wall was kept 10 Å. Energy of the prepared systems was minimized for 5000 steps using the steepest descent method or until a gradient threshold of 25 kcal/mol/Å was achieved. It was followed by L-BFGS (low-memory Broyden–Fletcher–Goldfarb Shanno quasi-Newtonian minimizer) until a convergence threshold of 1 kcal/mol/Å was met. Further, the minimized systems were equilibrated in mainly 5 steps in NVT and NPT ensembles using a “relax model system” before the simulation option in the Desmond Schrodinger suite. In the “relax model system” by default there are 5 equilibration steps, the first two steps of the equilibration are performed in the NVT ensemble at 10 K temperature for 100 and 12 ps, respectively, with restraints on solute heavy atoms in the next three steps of equilibration. NPT ensembles are used for 12 ps (with restraints) at 10 K, 12 ps (with restraints) at 300 K, and 24 ps (without restraints) at 300 K, respectively. The equilibrated systems were then subjected to 30 ns unrestrained MD simulations in the NPT ensemble with 300 K temperature maintained by Nose–Hoover chain thermostat, constant pressure of 1 atm maintained by Martyna–Tobias–Kelin barostat, and a integration time step of 2 fs [23,25].

### 2.3. Analysis of the MD Simulations

The MM/GBSA (molecular mechanics energies combined with the generalized Born and surface area continuum solvation) free binding energy was calculated using the prime module of the Schrodinger suite. The last 10 ns from each of the trajectory was used for this computation using the following equation:MM/GBSA ΔG_bind_ = ΔG_complex_ − (Δ G_receptor_ + Δ G_ligand_)ΔG = ΔE_gas_ + ΔG_sol_ − TΔS_gas_ΔE_gas_ = ΔE_int_ + ΔE_elec_ + ΔE_vdw_ΔG_sol_ = ΔG_gb_ + ΔG_surf_(1)

The prime module was used to compute all the energy components using the coordinates of complex, receptor and ligand using OPL3e forcefield. The binding free energy (ΔG_bind_) is dissociated in binding free energy of the complex, receptor and ligand. The gas-phase interaction energy (ΔE_gas_) is calculated as the sum of electrostatic (ΔE_elec_) and van der waal (ΔE_vdw_) interaction energies, while internal energy was neglected. The solvation free energy (ΔG_sol_) contains non-polar (ΔG_surf_) and polar solvation energy (ΔG_gb_) that is calculated by using VSGB solvation model and OPL3e force field, while the entropy term is neglected by default. The hydrogen bond interaction occupancy was calculated using Visual Molecular Dynamics (VMD) [26]. The polar and non-polar interactions of the protein–protein interface from the averages structures of the simulated complex was analyzed using LigPlot [27].

## 3. Results

### 3.1. Structural Impact of Mutations Occurring in S Protein 

Mutations occurring in either N-terminal domain (NTD) or receptor binding domain (RBD, which initiates contact with host receptors) of the S protein can contribute to host tropism (Table 1). The H69, V70 and Y144 deletions were observed to be localized on solvent-accessible β-hairpin loops in the NTD (Figure 1D). In the case of murine coronavirus S protein, its NTD was associated with an extended host range of viruses. These deletions originally observed during transmission in the mink population also highlight adaptation possibilities of SARS-CoV-2 in susceptible animal reservoirs [11]. In two recent pioneering studies, monoclonal antibodies isolated from convalescent COVID-19 patients were found to interact specifically with the NTD of the S protein, reinforcing that the mutations in the solvent-exposed epitope regions could confer antibody resistance [28,29]. Besides NTD deletions, the most concerning mutation common to 501Y.V1, 501Y.V2 and P.1 lineages is N501Y, located in the RBD region (Figure 1D). The highly flexible RBD of the SARS-CoV-2 S protein initiates its contact with host ACE2 receptors [6,30,31], is dependent on glycation of N165 and N234 [32], and is an attractive target for several neutralizing antibodies [33,34,35]. The N501Y mutation can promote S protein affinity with host ACE2 receptors [17]. Mutation analysis revealed a stronger interaction network of Y501 compared to wildtype N501 (Figure 1D and Appendix A). The N501Y mutation was, however, distantly located from the CR3022/C135 (SARS-CoV-2 neutralizing monoclonal antibodies)–RBD interaction interface (Appendix A) [36]. However, both antibodies share distinct epitope regions in the RBD of the S protein (Appendix A). The N501Y was also observed to have highest co-occurrence (>90%) with R203K, G204R mutations in the N protein, widely accumulated in North American and European populations (Table 1) [37,38]. The 501Y.V2 and P.1 variants show a different set of mutations besides N501Y—E484K in both and K417N (in 501Y.V2) and K417T (in P.1). The N501Y, E484K, K417N/T mutations are localized near the ACE2 interaction interface but distant from C135/CR3022 binding sites (Appendix A). Apart from these two lineages, a recent analysis (preprint) identified emerging variants of SARS-CoV-2 in Australian and Indian populations which can evade host immune responses [39]. The highly frequent S477N mutation in the Australian population was located outside the RBD–C135/CR3022 interaction interface (Appendix A) [39]. However, in India, the N440K high frequency variant was observed to be located at the C135–RBD interaction interface [39], where the mutation led to a weaker interaction network (Appendix A). This could also possibly explain immune evasion by this variant during asymptomatic reinfection in two healthcare workers [40]. Interestingly, the N440K mutation was the only mutation to show 100% co-occurrence with C64F in the membrane glycoprotein of SARS-CoV-2 (Appendix A), along with the globally frequent D614G (S protein) and P323L (ORF1ab) variants [4,16]. Apart from deletions in the NTD and mutations in the RBD region, the 501Y.V1 harbors P681H mutation in the vicinity of the polybasic ‘furin (PRRAR)’ cleavage site [9,13,17]. SARS-CoV-2 uses host factors such as transmembrane protease serine 2 (TMPRSS2) for priming of the S protein and mediating virus–host membrane fusion [41]. Protein–protein interaction analysis showed that the P681H mutation can putatively promote association of cleavage sites with TMPRSS2 (Appendix A). Recent studies have indicated selection pressure in favor of this mutation with an exponential increase in sequences (>50 k) isolated from the USA and UK [42]. Loss of Pro in vicinity of fusion sites can alter infectivity, fusion kinetics and pathology of mouse hepatitis virus [43].

### 3.2. MD Simulation Analysis of S Protein with ACE2 Receptors

MD simulation outcomes of S protein–ACE2 complexes showed better H-bond networks of N440K (3.89 ± 1.64) compared to wildtype spike (3.83 ± 1.64) and N501Y (3.30 ± 1.31) throughout the simulation period (Figure 2A). During simulations, in wildtype S protein, residues K417, Y449, Y489, Q493, T500 and G502 were involved in hydrogen bonding with ACE2. In the case of N501Y and N440K, the interactions were identical to the wildtype, however two crucial interactions, T500 and G502 were lost in N501Y variant (Appendix A). When the average structure from the simulated trajectories was analysed, it was found that the 501st residue of the spike protein was involved in hydrophobic interactions only in the wildtype and N501Y mutant (Figure 2B,C). MM/GBSA analysis of binding energies showed highest binding affinity of wildtype (−103.97 ± 14.01 kcal/mol). The S protein was observed to have the highest binding affinity towards ACE-2, followed by the N440K mutant (−93.14 ± 16.68 kcal/mol) and N501Y (−89.80 ± 7.30 kcal/mol) (Figure 2D).

### 3.3. Docking and MD Simulation Analysis of RBD Mutants with C135 Antibody

Protein–protein docking and refinement showed apparent decrease in global interaction energy with C135 antibody on introduction of different mutation clusters in the RBD of the S protein (Appendix A). During MD simulations, decent H-bond networks were observed for interaction of RBD with the heavy chain of the C135 antibody. However, the mutants (N440K and N501Y) had the highest number of hydrogen bond contacts in most of the frames (Figure 3A). When hydrogen bond occupancy was checked, it was found that in case of wildtype, 12 residues of the spike protein were involved in the hydrogen bonding. Among all of them, Y449 made significant interaction with H100 of around 0.31 fraction of time. In the case of N501Y, total number of interactive residues increased to 20, and among these, N440 and N442 were best and they were interacting with F52 and S95 for 0.11 and 0.17 fraction of time, respectively. Finally, in the case of N440K, the total number of hydrogen bond interactive residues was found to be 12. Among the 12 residues, Q398 was interacting with S31 and S96 for 0.14 fraction of simulation time, while L441 with Y98 for 0.08 fraction of simulation time and mutated residue K440 was interacting with N56 for 0.03 fraction of simulation time. The crucial interactions are depicted through the average structures extracted from the MD simulations in Figure 3B–D. Finally, in terms of binding energy, the highest affinity of C135 was found with N440K (−45.03 ± 12.14 kcal/mol) followed by N501Y (−28.63 ± 16.20 kcal/mol) and wildtype (−25.86 ± 13.10 kcal/mol).

### 3.4. Docking and MD Simulation Analysis of RBD Mutants with CR3022 Antibody

Protein–protein docking outcomes for wildtype and mutant RBD with CR3022 antibody showed a similar decrease in global interaction energy upon accumulating different mutations (Appendix A). In the case of RBD–CR3022 interactions analysed during MD simulations, the number of hydrogen bond counts suggested that the average hydrogen bond interaction between N501Y mutant (3.32 ± 1.2) and CR3022 significantly reduced in comparison to wildtype (5.63 ± 2.32) and N440K (5.90 ± 2.14) (Figure 4A). When the hydrogen bonds throughout the simulation were checked, it was found that G381, K386 and R408 of the spike protein were major interacting residues with 41 and 79 percent of the interaction time. In case of the N501Y mutant, the residues K378, G381 and K386 were the major interacting ones, having interaction for more than 50% of the simulation time. Similarly, in case of the N440K mutant, the residues K381, G386 and R403 were the major interactive residues. The polar and non-polar interactions of spike and CR3022 of the average structures are shown in Figure 4B–D. In line with hydrogen bond counts, the MM/GBSA binding free energy showed that the binding affinity of N501Y mutant (−90.03 ± 10.08 kcal/mol) decreased in comparison to the wildtype (−103.01 ± 23.39 kcal/mol) and N440K mutant (−100.53 ± 8.76 kcal/mol) towards the CR3022 antibody. Taken together, our data suggest that the naturally occurring mutations in SARS-CoV-2 virus are likely be implicated in variability in binding to neutralizing antibodies.

### 3.5. Structural Impact of Mutations Occurring in ORF8 Protein

The other mutations which differentiate 501Y.V1 and 501Y.V2, P.1 variants are localized in ORF8 and E proteins (Table 1). The ORF8 protein is proposed to interact with variety of host proteins and modulate immune responses, and disrupt IFN-I signaling and downregulate expression of MHC-I in cells [44,45]. The R52I and Y73C mutations in ORF8 of 501Y.V1 variant are localized at its dimerization interface. In the wildtype strain, R52 forms H-bond contact with I121 (Appendix A) while the Y73 forms part of crucial _73_YIDI_76_ non-covalent ORF8 dimer interaction interface, unique to SARS-CoV-2. The presence of these interfaces enables ORF8 to form large-scale assemblies, possibly aiding SARS-CoV-2 to evade and modulate host immune responses [44]. The P.1 specific E92K in ORF8 showed >90% co-occurrence with its L84S mutation and S202N, M86I in N and nsp6 (ORF1ab) proteins, respectively. 

### 3.6. MD Simulation Analysis of ORF8 Dimer Interface

The R52I localized at the dimer interface could affect dimer assembly process of ORF8. Accordingly, we wanted to check if this mutation leads to an increase/decrease in interaction of two monomers, which could in turn affect higher order assembly process of ORF8 proteins. Simulations of R52I variant showed slight lowering of interaction affinity between ORF8 monomers compared to wildtype strain. When the simulated ORF8 proteins were investigated for the number of hydrogen bonds, it was found that both wildtype ORF8 (6.16 ± 1.77) and R52I (6.20 ± 1.81) had a similar number of hydrogen bonds with another protomer throughout the simulations (Figure 5A). Further, when the simulated trajectories were analysed for investigating the hydrogen bond occupancy of R52 of wildtype, it was found that it was making 0.11 fraction with GLN27 of another protomer while no significant hydrogen bond interaction was found when ARG52 was mutated to I52 (Appendix A). To check the other crucial interactions between the protomers near the 52nd residue of ORF8, the average structure from the simulated trajectories was analysed. In the average structure, it was observed that R52 was making hydrophobic interactions with Q27, while Q18 and K53 were making hydrogen bonds (Figure 5B). Similarly, I52 was also involved in the hydrophobic interaction with another protomer (Figure 5C). Finally, when MM/GBSA free binding energy was calculated between the protomers, it found that wildtype (−43.92 ± 12.07 kcal/mol) had slightly better binding than R52I (−39.90 ± 12.29 kcal/mol) (Figure 5D).

### 3.7. Structural Impact of Mutations in E and N Proteins

The P71L mutation in the E protein (501Y.V2 variant) is located in the vicinity of the putative host MPP5-interacting C-terminal domain but was not observed to perturb local interaction network (Appendix A). The SARS-CoV-2 N protein can exist both in monomeric and oligomeric forms, and can interact with RNA. The dimerization interface is formed through interactions of C-terminal domain (256–364, PDB id: 6wzo) while the RNA interactions are mediated through its N-terminal domain (46–176, PDB id: 7acs) (Appendix A). Recently, it was shown that ORF9b (alternative ORF within the N gene) can suppress IFN-I responses through physical interactions with mitochondrial TOM70 or induction of lactic acid production [46]. The D3L and T205I, S235F mutations in the N protein occur outside the dimer and RNA interaction interfaces, in the unstructured regions in its NTD and linker regions, respectively (Appendix A). However, mutation analysis predicted highest stabilization effect conferred by S235F to the N protein (Appendix A). 

Protein stability predictions assessed from the impact of mutations showed mild stabilization of the S protein RBD by N501Y and destabilization by the E484K mutation. The effects of other mutations are detailed in Appendix A. The presence of an entirely different set of co-occurring mutations with each mutation in 501Y.V1, 501Y.V2 and P.1 variants (Table 1) could likely affect transmissibility and infectivity of the virus. From a drug targeting perspective, the K417, N440, S477, E484 and N501 residues form part of key residues participating in RBD–drug interactions (Appendix A). Consequently, the mutations are likely to affect proposed drug interactions inside RBD-binding pockets. 

## 4. Discussions

Impact of mutant variants on clinical, diagnostic, therapeutic and public health strategies.

Considering the structure–function impact of mutations, emergence of these new variants is of urgent concern; it would likely impact key functionalities associated with COVID-19 infection including transmissibility, disease severity, diagnostic sensitivity and specificity, and vaccine-induced protection. In likely setbacks caused by the mutant strains, 80% tend to develop milder symptoms that escape early clinical screening of fever, cough, upper respiratory infections, etc., while the remaining 20% presented with severe symptoms of ‘cytokine storm’ leading to re-occupation of ICU and hospital beds once again. In a latest pioneering analysis of disease severity in SARS-CoV-2-infected individuals, higher ratios of anti-RBD IgG antibodies (compared to antibodies against N protein) were observed for mildly ill patients [47]. In critically ill patients, antibodies recognizing viral proteins other than S were observed. This raise concerns on disease severity upon reinfection with strains accumulating clusters of mutations (as observed in 501Y.V1, 501Y.V2 and P.1 lineages), indicating that antibody-based, and other diagnostic tests adopted in various surveillance programs for prior detection of virus might also result in underestimating the breadth of the pandemic (Appendix A). The sensitive PCR tests are likely to miss mutant antigen and likely to cause a missed diagnosis [48]. A single nucleotide change can alter the primer/probe binding site on the target gene, as reported by Vanaerschot et al. on reduced sensitivity of the RT-PCR diagnostic assay owing to a single mutation (Q289H) in the forward N gene primer [49]. 

The S protein and its RBD is a promising vaccine target against SARS-CoV-2, in particular, its RBD which interacts with the host cellular receptor to gain entry inside the host cells. The neutralizing antibodies in sera predominantly target RBD domain motifs, mutations in which could significantly impact binding and neutralization (Table 1). Currently, eight different vaccine types are in various stages of clinical trials (Appendix A). The COVID-19 vaccine type to be used in the regions of emerging variant strains will be a crucial factor in determining the efficacy of vaccine-induced immunity. S gene or protein-based stand-alone vaccines might need to be tweaked a little to continue to be effective. Available vaccines based on encoded mRNA or vector-linked spike glycoproteins produce antibodies specific to wildtype or highly frequent variants. However, the accumulation of multiple mutations in newly emerging variants may bypass the immune recognition by the vaccine-induced antibodies. Although the Moderna COVID-19 vaccine (mRNA 1273) could be effective against the N501Y mutation, additional mutations in the South African variant (501Y.V2), Indian and Australian variants are likely to influence its S protein affinity to the host receptor and evading neutralization. Since the start of the pandemic, researchers have identified thousands of viral mutations which could hinder the activity of potent neutralizing antibodies and block the virus invasion of target cells. Among the neutralizing antibodies that the mutation obstructed were those in the blood of people who had recovered from COVID-19, as well as some manufactured monoclonal antibodies that are being developed into treatments. The mutant coronavirus will tend to give slip to designer antibodies that are manufactured for its treatment [50], however, there is no peer reviewed experimental evidence to show that the new SARS-CoV-2 variants would escape vaccine-induced neutralizing antibodies or manufactured therapeutics [51]. Given the time-consuming process of licensing a vaccine along with the emergence of new variants, immunization programs might become perennial till the pandemic subsides. This might compel vaccine manufacturers to design updated candidates and roll out of recurring “flu-like shots” annually for tackling COVID-19.

Globally, out of 291 candidate vaccines in various stages of development, most vaccines will require two doses given 3–4 weeks apart (COVID-19 vaccine landscape). In a country like India, which has the second largest infection burden globally, the priority groups in India who will be offered COVID-19 vaccines include the following—4.7 million healthcare workers, 50 million frontline workers in the police, defense, railways, hospitality industry as workforce, and 260 million citizens above the age of 50 years. The latter group includes 180 million people aged 50–60 years and the remaining 80 million people aged 60-plus years (https://www.statista.com/topics/754/india/, accessed on 4 January 2021). Successful introduction of a vaccine will largely depend on the quality of training given to the personnel involved in the process. It will be modeled just like the election process, and include the following four components, namely—safety precautions comprising of adverse event following immunization (AEFI), management of the supply and cold chains, community participation and communication strategy, and monitoring and supportive supervision (MoHFW, India). While the National Disaster Management agencies in all countries will keep a strict vigil on surveillance and monitoring of vaccine delivery in priority groups, with an eye on vaccine-related adverse effects, the likely fresh reinfections with emerging VOC(s) and their outcomes in vaccinated individuals will be a matter of concern. Recent publications related to the spread of 501Y.V1 and P.1 variant in Manaus, Brazil and the Y501.V2 variant in South Africa are providing credible evidence of reduced protection of first generation vaccines against the emerging variants [52,53]. On the other hand, fresh studies on BNT162b2 have shown promising neutralization antibody geometric mean titer against the 501Y.V1 and 501.V2 variants [54].

## 5. Conclusions 

The emergence of new 501Y.V1, 501Y.V2 and P.1 variants marks the beginning of antigenic drift for SARS-CoV-2 owing to cluster of mutations acquired in the S protein [55]. The rise of such variants is particularly concerning as these have might escape antibody therapies and vaccine induced protection, during a period when massive vaccination drives are already in progress globally. As shown here, the N501Y mutation in all three variants may enhance ACE2 affinity but might not confer antibody resistance individually [56] or neutralizing effects by convalescent plasma and vaccine sera [54]. In accordance with recently published reports, our findings also indicate reinfection potential of 501Y.V2 and P.1 variants in the South African and Brazilian population, respectively [52,57]. In the 501Y.V1 variant, we hypothesized that R52I mutation could affect multimer assembly of the ORF8 protein, however another Q27Stop deletion apparently showed no effect on viral transcription [58]. Mutations in the N and E proteins occur outside RNA and protein interaction interfaces, while their exact role in viral pathogenesis still needs to be deciphered. In a nutshell, the cluster of mutations in the S protein can have significant impact on viral transmission, infectivity, diagnostics and host immune responses. The N501Y mutation, in particular, leads to stronger interaction with human ACE2 compared to its wildtype, although other co-occurring mutations along with N501Y might have a different overall effect. In addition, the combined impact of all the mutations in the variants warrants further studies to provide insights into the infectivity and pathogenesis associated with the variants. 

The advent of new immune-escaping strains can further shift decline of the pandemic and continue to disrupt global health services [59]. According to latest modeling studies, the added strains from the pandemic can also significantly escalate deaths from other infectious diseases [60]. For example, emergence of the new variants in India, South Africa and their adjoining nations could be alarming, as these are already among top contributors accounting to 87% of new tuberculosis cases in 2019 (WHO TB-Fact Sheet 2020, Accessed: 3 January 2021). While the mutations could also have significant impact on diagnostic assays owing to S gene target failures, it will likely have an overall clinical, therapeutic and public health impact and post-vaccine reinfections with new variants are likely to keep everyone guessing for a long time to come!

## Figures and Tables

**Figure 1 viruses-13-00439-f001:**
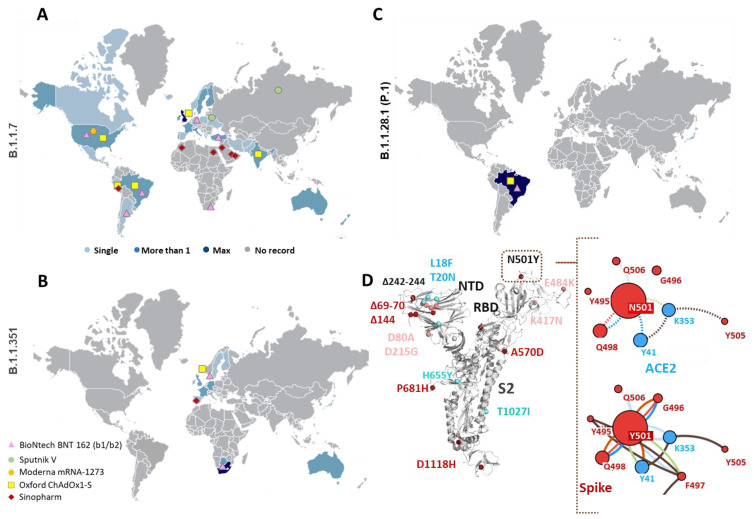
Emergence of new B.1.1.7, B.1.351 and P.1 variant lineages. (**A**–**C**) Global distribution of sequences arising from various nations reporting 501Y.V1 (B.1.1.7), 501Y.V2 (B.1.351) and P.1 (B.1.1.28.1) variants, respectively. Pinned colored shapes on the map indicate major vaccine trials in various regions around the globe. Geographical pinning of vaccines in regions with rising frequency of variant population indicate the need to re-assess these candidates against new variants. Single (Light Blue), More than 1 (Blue) and Max (Dark Blue) indicate number of sequences of specific variants originating from different nations. (**D**) Structural mapping of mutations from 501Y.V1 (dark red dots) and 501Y.V2 (pink dots) variants on the spike (S) protein of Severe Acute Respiratory Syndrome-Coronavirus 2 (SARS-CoV-2). The amino acid deletions (Δ marked) in both variants are located in the N-terminal domain (NTD). The common N501Y mutation is located in receptor binding domain (RBD) region which makes contact with host angiotensin II converting enzyme (ACE2) receptors. Residue interactions of N501 (dashed lines) and Y501 (solid lines) with ACE2 (Blue dots) and other residues of the S protein (red dots) are shown in the right panel. Structural analysis of the mutations shows higher interaction network in Y501-ACE2 compared to wildtype N501-ACE2. Color codes—H-bonds (red), polar H-bonds (orange), VdW (light blue), aromatic (light green) and ring–ring interactions (brown).

**Figure 2 viruses-13-00439-f002:**
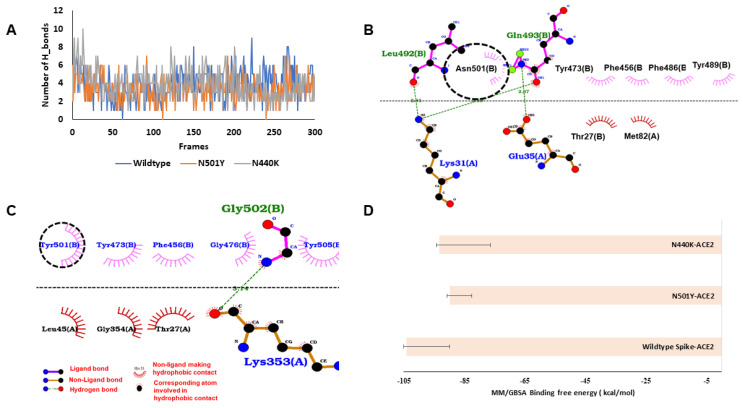
Binding interactions and energy of Spike proteins with ACE-2. (**A**) The number of hydrogen bonds between the wildtype and mutant (N501Y and N440K) spike protein and ACE2 during the simulation. (**B**) Wildtype spike–ACE2 interactions in the average structure extracted from MD simulations, N501 (circled) making hydrophobic contact (hydrogen bonds are shown with green dots and non-polar interactions with magenta and brick semicircle). (**C**) N501Y–ACE2 interactions in the average structure. (**D**) Molecular mechanics energies combined with the generalized Born and surface area continuum solvation (MM/GBSA) binding free energy of spike proteins with ACE2.

**Figure 3 viruses-13-00439-f003:**
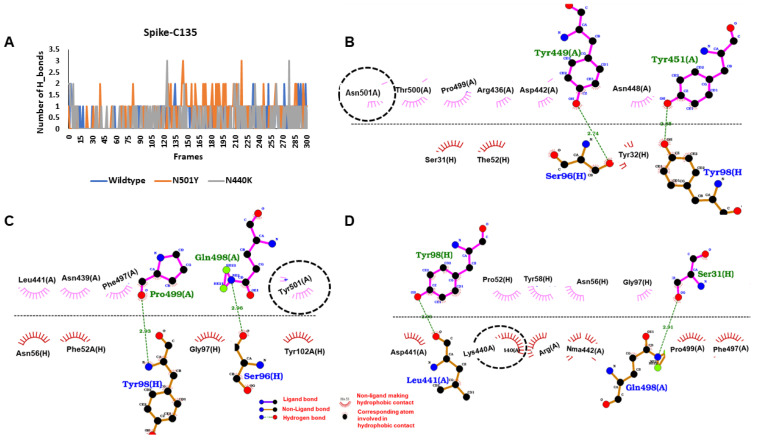
Binding interactions and energy of spike proteins with C135 antibody. (**A**) The hydrogen bond peaks showing the interaction of mutants were better than wildtype spike protein. (**B**) The binding interaction of wildtype spike protein with heavy chain of C135 antibody, N501 (circled) making hydrophobic contact (hydrogen bonds are shown with green dots and non-polar interactions with magenta and brick semicircle). (**C**) The binding interaction of N501Y mutant spike protein with heavy chain of C135 antibody. (**D**) The binding interaction of N440K mutant spike protein with heavy chain of C135 antibody.

**Figure 4 viruses-13-00439-f004:**
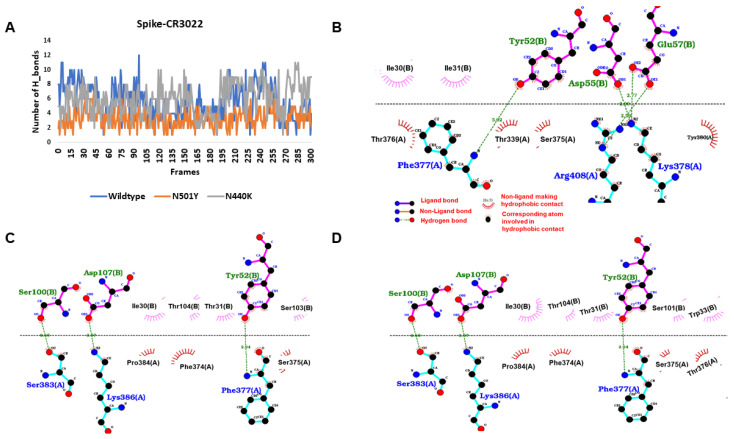
Binding interactions and energy of spike proteins with the CR3022 antibody. (**A**) The hydrogen interaction of wildtype was better than mutant spike proteins. (**B**) The binding interaction of wildtype spike protein with the heavy chain of CR3022 antibody, N501 (circled) making hydrophobic contact (hydrogen bonds are shown with green dots and non-polar interactions with magenta and brick semicircle). (**C**) The binding interaction of N501Y mutant spike protein with heavy chain of CR30222 antibody. (**D**) The binding interaction of N440K mutant spike protein with heavy chain of CR3022 antibody.

**Figure 5 viruses-13-00439-f005:**
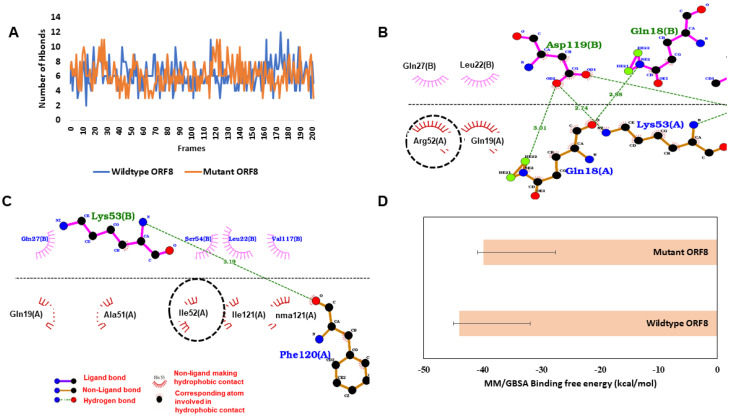
Binding affinity between ORF8 protomers. (**A**) The number of hydrogen bonds between the wildtype (WT) and mutant (MT) ORF8 protomers during the simulation. (**B**) Wildtype ORF8 interactions in the average structure extracted from MD simulations, R52 (circled) making hydrophobic contact (hydrogen bonds are shown with green dots and non-polar interactions with magenta and brick semicircle). (**C**) Mutant ORF8 interactions in the average structure. (**D**) MM/GBSA binding free energy in kcal/mol.

**Table 1 viruses-13-00439-t001:** Notable non-synonymous mutations and deletions (Δ) in S, envelope (E), ORF8, nucleocapsid (N) proteins of SARS-CoV-2 501Y.V1, 501Y.V2 (*) and P.1 (^#^) variants. Co-occurring mutations other than D614G (S protein) and P323L (ORF1ab) are shown here. Protein names are shown in parenthesis. Table adapted from Portelli et al. and the Public Health England report [16,17]. References for modulation in hACE2 affinity or antibody resistance (shown as superscripts) are provided in Appendix A.

Gene	Mutation	hACE2 Affinity	Antibody Resistance	Global Frequency (%)	Co-Occurrence > 90%	Co-Occurrence 50–90%
	D80A *			0.02	P679L (ORF1ab), Q57H (ORF3a)	Y153C (ORF1ab)
S	ΔH69 ΔV70 ΔY144	-	Increase ^S13–14^ (Weak binding of ΔY144 to A48 antibody)	-		
	D215G *	-		0.04	A46V (ORF1ab), Q57H (ORF3a)	S6L (ORF1ab)
	K417N *	Minimal ^S19^	Increase to A48 antibody ^S17^	<0.01	-	-
	K417T ^#^					
	E484K *	Increase ^S19^	Increase ^S18^ (Weak binding with C121 or C144 antibody)	<0.01	-	F70C, L353F, T428I, G15S (ORF1ab), R203K, G204R (N)
	N501Y	Increase^S15^	Increase ^S15^ (Weak affinity with STE90-C11 antibody)	<0.01	R203K, G204R (N)	-
	A570D	-	-	<0.01	-	T1384S (ORF1ab)
	H655Y ^#^			0.04	-	-
	P681H	-	-	<0.01	-	-
	A701V *	-	-	<0.01	-	-
	T716I	-	-	0.03	-	R203K, G204R (N)
	S982A	-	-	<0.01	-	
	T1027I			0.02	P323L (ORF1ab), D614G (S)	R203K, G204R (N)
	D1118H	-	-	<0.01	-	
	S477N (Australia)	Increase ^S16^	Increase ^S17^ (Weak binding to Fab 2–4)	0.14	R203K, G204R (N)	-
	N440K (India)	-	Increase (Weak binding with C135 antibody)	<0.01	C64F (Membrane glycoprotein)	-
ORF8	Q27Stop	-		-		
	R52I	-		<0.01	-	R203K, G204R (N)
	Y73C	-		<0.01	-	-
	E92K ^#^	-	-	0.02	L84S (ORF8), S202N (N), M86I (ORF1ab)	
N	D3L	-		0.02	R203K, G204R (N)	L864V (S)
	S235F			0.01	D144A, K384N (ORF1b)	Q19H, P503S, P985S, L1706F (ORF1ab)
	T205I *	-	-	0.14	-	-
	P80R ^#^	-	-	<0.01	-	T85I, P323L, D484A (ORF1ab), D614G, D936Y (S), K16N, Q57H (ORF3a)
E	P71L *	-	-	0.04	-	-

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
