# Peer review of "Structure-Function Analyses of New SARS-CoV-2 Variants B.1.1.7, B.1.351 and B.1.1.28.1: Clinical, Diagnostic, Therapeutic and Public Health Implications"

_viruses, 2021, doi:10.3390/v13030439_

Round 1
Reviewer 1 Report
The article by Singh et al. represent to my knowledge one of the first studies to address the molecular consequences of the new three variants of concerns (VOC): B.1.1.7, B.1.135 and B.1.1.28 known as the UK, South African and Brazilian variants. The study relate to specific cases, the spike (S), the ORF8 and nucleocapsid (N) proteins. The article provides a link between mutations and possible changes in the protein structure. I consider the article perse interesting in a hot topic (SARS-CoV-2 mutations) nowadays, however I think the authors need to make a big effort to bring a high quality manuscript which not only describe the story, but also captures audience and show high quality data. I will be glad to see my comments implemented before accepting it for publication.
Comments:
1) Since, the manuscript (MS) spent quite certain amount of time in the spike (S) protein, I suggest to have a section or at least describing the structure and stability of the S protein (e.g. S1, S2, S1/S2), conformations (closed and open) and the role of glycans in supporting the transition from RBD down to up prior to ACE2 receptor recognition. I believe the authors could include some recent references about the stability and conformation (https://doi.org/10.1039/D0NR03969A, and this published recently MDPI https://doi.org/10.3390/ma13235362). In addition, about the role of glycans beyond shielding (https://doi.org/10.1021/acscentsci.0c01056).
2) Line 44: citation for original study by Korber et al. (https://doi.org/10.1016/j.cell.2020.06.043) should be included.
3)Line 49,Line 58, Line 72, Line 82, Line 200-201 and so on.... I find quite confusing across the MS to see different nomenclatures for the new SARS-CoV-2 linages. The title uses one, the intro another and it keeps switching in different parts. Please, I suggest to set one nomenclature (as the one in the title). Then, briefly describe all the other, but use systematically one over the whole MS.
4) Line 56---> should be EuroMOMO
5) Line 52 with Line 55: check consistency in English
6) Line 60 and Line 70: Remove inculcates---> replace by "show, display.."
7) Line 90: In Figure 1, it is not explained the meaning of Single, More than 1, Max and No recorded. Same problem with variant nomemclature. Use one please.
8) Line 104: I disagree not to see the methodology, specially the molecular dynamics in the main text.
9) Line. 109, Line 11: check consistency in calling PDB files
10) Line 127-128: Mutation...to wild-type N501: To my knowledge it can include as part of the abstract as it is a finding of the study.
12) Line 134: again replace inculcate13) Line 149 to Line 162. The author call Figure S5 and next time Figure S10. This means there is missing citation to Figure S6-S9 or the authors confuse the citation sequence in the SM. Please check well before using in the main text
14) Line 170: missing the pronoun, "it".
15) Refrase the sentence in Line 155-156, not clear
16) Missing full stop in "et al." e.g. Line 184. Check in other parts
17) Line 181: Caption of Table 1. Missing the definition of "delta symbol"= deletions
18) Line 238-240: Please add a citation, otherwise remove it.
19) Line 241: What the author mean by " enumerators" ?
20) Line 250: Finally, the conclusions are not representative of the whole work. Please expand it more. In particular with emphasis on the N501 spike mutation and its consequences. Same for other mutation along the S protein/
Finally, I would suggest to decrease the length of the supplementary material and take method, analysis onto the main manuscript. If so, also increase the quality of the figures. So far, I can not read the names of residues, they are obscured by the contours of the protein.
Author Response
Reviewer 1
The article by Singh et al. represent to my knowledge one of the first studies to address the molecular consequences of the new three variants of concerns (VOC): B.1.1.7, B.1.135 and B.1.1.28 known as the UK, South African and Brazilian variants. The study relate to specific cases, the spike (S), the ORF8 and nucleocapsid (N) proteins. The article provides a link between mutations and possible changes in the protein structure. I consider the article perse interesting in a hot topic (SARS-CoV-2 mutations) nowadays, however I think the authors need to make a big effort to bring a high quality manuscript which not only describe the story, but also captures audience and show high quality data. I will be glad to see my comments implemented before accepting it for publication.
Comment 1: Since, the manuscript (MS) spent quite certain amount of time in the spike (S) protein, I suggest to have a section or at least describing the structure and stability of the S protein (e.g. S1, S2, S1/S2), conformations (closed and open) and the role of glycans in supporting the transition from RBD down to up prior to ACE2 receptor recognition. I believe the authors could include some recent references about the stability and conformation (https://doi.org/10.1039/D0NR03969A, and this published recently MDPI https://doi.org/10.3390/ma13235362). In addition, about the role of glycans beyond shielding (https://doi.org/10.1021/acscentsci.0c01056).
Response: We thank the Reviewer for suggesting the above-mentioned research articles. We have described the structure and stability of the S Protein and included these references in the manuscript (Page 5, Lines 25-26).
Comment 2: Line 44: citation for original study by Korber et al. (https://doi.org/10.1016/j.cell.2020.06.043) should be included.
Response: We are thankful to the Reviewer for this suggestion. The study by Korber et al. has been included in the manuscript (Page 2, Line28)
Comment 3: Line 49,Line 58, Line 72, Line 82, Line 200-201 and so on.... I find quite confusing across the MS to see different nomenclatures for the new SARS-CoV-2 linages. The title uses one, the intro another and it keeps switching in different parts. Please, I suggest to set one nomenclature (as the one in the title). Then, briefly describe all the other, but use systematically one over the whole MS.
Response: We appreciate the Reviewer’s comment. We have edited the manuscript accordingly and have used one set of nomenclature throughout. (Page 2, Line 31; Page 3, Line 11, 20 and 29; Page 5, Line 29; Page 9, Line 34)
Comment 4: Line 56---> should be EuroMOMO
Response: We have edited the manuscript accordingly (Page 3 , Line 2)
Comment 5: Line 52 with Line 55: check consistency in English
Response: We appreciate the Reviewer’s comment. We have edited the manuscript accordingly. (Page 3, Lines 1-4).
Comment 6: Line 60 and Line 70: Remove inculcates---> replace by "show, display."
Response: The authors are thankful to the Reviewer’s suggestion. We have edited the manuscript according to the suggestion. (Page 3 , Line 10 and 19; Page 6, Line 4 )).
Comment 7: Line 90: In Figure 1, it is not explained the meaning of Single, More than 1, Max and No recorded. Same problem with variant nomenclature. Use one please.
Response: We regret this error. Single, More than 1, Max indicate number of sequences of specific variants originating from different nations. We have included the meaning of these in Figure 1 legend.
Comment 8: Line 104: I disagree not to see the methodology, specially the molecular dynamics in the main text.
Response: As per Reviewer suggestion, we have now included full methodology section into the main manuscript.
Comment 9: Line. 109, Line 11: check consistency in calling PDB files
Response: We have checked calling of PDB files and this has been fixed in all parts of the manuscript.
Comment 10: Line 127-128: Mutation...to wild-type N501: To my knowledge it can include as part of the abstract as it is a finding of the study.
Response: As per Reviewer suggestion we have now included N501Y mutation in the Abstract section.
Comment 11: ??
Response: The Reviewer has not made Comments and moved to Comment 12.
Comment 12: Line 134: again replace inculcate.
Response: As per Reviewer suggestion, we have replaced the word “inculcate” with “showed/displayed” everywhere in the text.
Comment 13: Line 149 to Line 162. The author call Figure S5 and next time Figure S10. This means there is missing citation to Figure S6-S9 or the authors confuse the citation sequence in the SM. Please check well before using in the main text
Response: Earlier, the figures were numbered based on their appearance in Main text and Supplementary text. Now we have move supplementary section into main text, thus Figures numbering has also been fixed accordingly.
Comment 14: Line 170: missing the pronoun, "it".
Response: We have made the suggested modification in the manuscript.
Comment 15: Refrase the sentence in Line 155-156, not clear
Response: We appreciate the Reviewer’s comment. We have rephrased the sentence accordingly.
(Page 8, Lines 1-2 )
Comment 16: Missing full stop in "et al." e.g. Line 184. Check in other parts.
Response: We have checked for these errors and edited the manuscript accordingly.
Comment 17: Line 181: Caption of Table 1. Missing the definition of "delta symbol"= deletions
Response: The authors have edited the manuscript accordingly. (Table No 1 legend)
Comment 18: Line 238-240: Please add a citation, otherwise remove it.
Response: As per Reviewer suggestion, we have added citation to this line.
Comment 19: Line 241: What the author mean by " enumerators" ?
Response: We have replaced this word with personnel.
Comment 20: Line 250: Finally, the conclusions are not representative of the whole work. Please expand it more. In particular with emphasis on the N501 spike mutation and its consequences. Same for other mutation along the S protein/
Response: We thank the Reviewer for the suggestion. We have now expanded the Conclusions section representing work done in the manuscript.
Comment 21: Finally, I would suggest to decrease the length of the supplementary material and take method, analysis onto the main manuscript. If so, also increase the quality of the figures. So far, I can not read the names of residues, they are obscured by the contours of the protein.
Response: As per Reviewer suggestion, we have moved major portion of Supplementary Section into main text and modified the quality of figures accordingly for better legibility.
Reviewer 2 Report
Singh et al. present a comprehensive analysis of SARS-CoV-2 mutations and their effects on predicted structural and functional characteristics of viral proteins. Overall the amount and scope of the results is adequate, although I cannot speak to the details of the structure-prediction methods used. The main weakness of the manuscript is its structure. Most of the original results are included in supplemental data. The key results and corresponding figures should be moved to the main text. The Result part of the text should be structured clearly to give it a more logical and clear structure. Current Figure 1 shows mainly the geographical spread of the mutant viurs lineages, which is not original result of this work and would be more appropriate in a review-style manuscript.
Author Response
Comment 1: Singh et al. present a comprehensive analysis of SARS-CoV-2 mutations and their effects on predicted structural and functional characteristics of viral proteins. Overall the amount and scope of the results is adequate, although I cannot speak to the details of the structure-prediction methods used. The main weakness of the manuscript is its structure. Most of the original results are included in supplemental data. The key results and corresponding figures should be moved to the main text. The Result part of the text should be structured clearly to give it a more logical and clear structure. Current Figure 1 shows mainly the geographical spread of the mutant viurs lineages, which is not original result of this work and would be more appropriate in a review-style manuscript.
Response: We thank the Reviewer for critical analysis of our manuscript and highlighting its weaknesses to further strengthen our work. As per Reviewer suggestions, we have restructured the manuscript by moving major part of Supplementary text and Figures into Main manuscript. The results part is also now structured for more clarity. We agree that Figure 1 shows geographical spread of mutant which is not our work. We had put this in introduction simply to bring the context in terms of spread of various mutant virus lineages globally. We prefer to retain this, giving appropriate citations, for the above reason.
Reviewer 3 Report
The manuscript by Singh was a relatively comprehensive research article on the potential roles of mutation forms of SARS-CoV-2 on the functional activities. The authors tested the bioinformatics-based predictions and protein structure-based functional analysis. Experiments were well designed. There were some moderate concerns:
- Rationale of the analysis on RBD was not clearly written, concerning the priming protein TMPRSS2 was demonstrated the importance.
- Lines 139-156 and related section can be added with a figure/table to summarize the ratio change /infectivity(if available) of each analyzed mutation form/strain.
- Missing docking analysis with neutralizing antibodies.
- The Conclusions section was not supported by the data.
Author Response
The manuscript by Singh was a relatively comprehensive research article on the potential roles of mutation forms of SARS-CoV-2 on the functional activities. The authors tested the bioinformatics-based predictions and protein structure-based functional analysis. Experiments were well designed. There were some moderate concerns:
Comment 1: Rationale of the analysis on RBD was not clearly written, concerning the priming protein TMPRSS2 was demonstrated the importance.
Response: We thank the Reviewer for the suggestion. We have incorporated more details on TMPRSS2 and the role of proline mutations in the vicinity of fusion sites (Page 6, Lines 13-16).
Comment 2: Lines 139-156 and related section can be added with a figure/table to summarize the ratio change /infectivity(if available) of each analyzed mutation form/strain.
Response: Although ratio in change in infectivity associated with each mutation is not available for specific antibodies, data/preprints in the form of antibody resistance associated with mutations have been appropriately referred in Table 1.
Comment 3: Missing docking analysis with neutralizing antibodies.
Response: We thank the Reviewer for this suggestion. We wish to highlight that we already performed MD simulations with some S protein RBD mutants and C135, CR3022 antibodies. However, as per Reviewer suggestion, we have performed docking analyses for all RBD mutants with C135 and CR3022 antibodies and detailed them as Table S1.
Comment 4: The Conclusions section was not supported by the data.
Response: We thank the Reviewer for the suggestion. We have now expanded the Conclusions section to comprehensively represent the work done and described in the manuscript.
Round 2
Reviewer 1 Report
I appreciate the effort, however, I still find some vague description. In particular the method section and figures. Before accepting for publication please address carefully this comments.
Line 21 page 4; replace Angstrom3 by Å. to the three
Line 22 page 4: citation to OPLS2 force field missing
Lin 23 page 4: replace " orthorhombic periodic boundary box" by orthorhombic box with periodic boundary conditions
Line 26 page 4. check the unit for force criterion in the steepest descent method. Author claim to be 25 kcal/mol/Å. I think it should be given in the range of 0.02-0.1 kcal/mol/Å (see https://manual.gromacs.org/documentation/current/reference-manual/algorithms/energy-minimization.html) under proper conversion from KJ/mol/nm
Line 28 page 4: "7 step" this is not clear, generally equilibrium simulations are run in ps-ns range. Please correct it
Line 19, page 7: replace K by k in Kcal and in other instances page 8,..
Lin 12 page 11: replace "neutralising geometric mean titers of antibodies" simply by neutralizing antibody geometric mean titer
Line 1-2 page 12: has to be rewritten, as it is not scientifically constructive
Overall quality of Figure 2-5 is still poor. Residues names, legend in figures, data in plots are not visible please do an effort to improve them.
Author Response
Reviewer-1
I appreciate the effort, however, I still find some vague description. In particular the method section and figures. Before accepting for publication please address carefully this comments.
Response: We thank the Reviewer for suggesting appropriate changes in the manuscript. Following are our point-wise responses to the comments.
Comment 1: Line 21 page 4; replace Angstrom3 by Å3.
Response: The change has been made in the revised manuscript (Page 4, Line 16).
Comment 2: Line 22 page 4: citation to OPLS2 force field missing
Response: The citation has been added in the revised manuscript (Reference No. 23).
Comment 3: Lin 23 page 4: replace " orthorhombic periodic boundary box" by orthorhombic box with periodic boundary conditions.
Response: As per Reviewer suggestion, the change has been made in the revised manuscript.
Comment 4: Line 26 page 4. check the unit for force criterion in the steepest descent method. Author claim to be 25 kcal/mol/Å. I think it should be given in the range of 0.02-0.1 kcal/mol/Å (see https://manual.gromacs.org/documentation/current/reference-manual/algorithms/energy-minimization.html) under proper conversion from KJ/mol/nm
Response: We understand the reviewer’s concern. The minimization was performed for 5000 steps and the minimization procedure was performed with steepest decent algorithm until the gradient threshold of 25kcal/mol/A was reached. This was followed by employing the LGFGS algorithm. However, the overall convergence was set to 1 kcal/mol/A. Further, the minimized system was extensively relaxed using the ‘relaxed system scheme’ before carrying out the production MD of the Schrodinger. This has been described in the revised manuscript as well as in the response to the next comment.
Comment 5: Line 28 page 4: "7 step" this is not clear, generally equilibrium simulations are run in ps-ns range. Please correct it.
Response: We understand the Reviewer’s concern. The manuscript has been revised by elaborating the method. The MD simulation was performed using Desmond module of Schrodinger software. In the Desmond module, the option of choosing a default relaxation protocol before performing the production simulation was used. This relaxation protocol consisted of 7 steps, out of which 5 steps were for equilibration. The first two steps of the equilibration steps were performed in NVT ensemble at 10 K temperature for 100 ps and 12 ps respectively, with restraints on solute heavy atoms. In the next three steps of equilibration, NPT ensemble were used for 12 ps (with restraints), 12 ps (with restraints), and 24 ps (without restraints) respectively.
Comment 6: Line 19, page 7: replace K by k in Kcal and in other instances page 8,..
Lin 12 page 11: replace "neutralising geometric mean titers of antibodies" simply by neutralizing antibody geometric mean titer.
Response: We thank the reviewer for the suggesting these changes. ‘kcal’ has now been uniformly used in the revised manuscript. Page 10, Line 30, "neutralising geometric mean titers of antibodies" is now replaced with neutralizing antibody geometric mean titer.
Comment 6: Line 1-2 page 12: has to be rewritten, as it is not scientifically constructive
Response: We understand the Reviewer concern. As per suggestion, we have rewritten the lines (Page 11, Lines 19-22) for clearer meaning.
Comment 7: Overall quality of Figure 2-5 is still poor. Residues names, legend in figures, data in plots are not visible please do an effort to improve them.
Response: The figures have been revised with larger fonts for improved clarity.
Reviewer 2 Report
The authors complied with my review comments. There seems to be some minor issue with figure formatting in the pdf.
Author Response
Reviewer-2
The authors complied with my review comments. There seems to be some minor issue with figure formatting in the pdf.
Response: The figures have been revised with larger fonts for the clarity. We will also upload figures separately so that they appear correctly in pdf print.
Reviewer 3 Report
The manuscript by Singh et al., was a relatively comprehensive research article on the potential roles of genetic mutations and their protein-protein interactions. The authors focused on potential influences in the antibody binding activities. The manuscript was significantly improved; however, there were some significant concerns:
- Page 8 Lines 7-18, the rationale of comparing ORF8 forms was unclear.
- Missing antibody type (such as IgM or IgG) information.
- Page 9 Lines 11-12 sentence can be followed with “indicating that”.
- Page 9 Line 21 “pathophysiology of the virus”was unclear.
- Page 12 Lines 1-4 was not based on data/references.
- Table 1 “Antibody resistance”was interesting but not well introduced or discussed.
- Labels barely readable in many of the figures.
Author Response
Reviewer-3
The manuscript by Singh et al., was a relatively comprehensive research article on the potential roles of genetic mutations and their protein-protein interactions. The authors focused on potential influences in the antibody binding activities. The manuscript was significantly improved; however, there were some significant concerns:
Comment 1: Page 8 Lines 7-18, the rationale of comparing ORF8 forms was unclear.
Reponse: We thank the Reviewer for the suggestion. The R52I mutation occurs at the dimerization interface of ORF8. We wanted to check if this mutation leads to increase/decrease in interaction of two monomers and accordingly alter higher assembly process of ORF8 proteins. As per Reviewer suggestion, we have also included rationale in the main manuscript (Page 8, Lines 6-8).
Comment 2: Missing antibody type (such as IgM or IgG) information.
Response: We have included antibody type information for CR3022 and C135 antibodies (Page 4, Line 8)
Comment 3: Page 9 Lines 11-12 sentence can be followed with “indicating that”.
Response: We have changed the structure of the sentence with addition of “indicating that” (Page 9, Line 17-21).
Comment 4: Page 9 Line 21 “pathophysiology of the virus”was unclear.
Response: We have changed “pathophysiology of the virus” to “transmissibility and infectivity” of the virus (Page 8, Line 38).
Comment 5: Page 12 Lines 1-4 was not based on data/references.
Response: We are not sure of the reviewers’ comment. Page 12, Lines 1-4 are references only.
Comment 6 : Table 1 “Antibody resistance” was interesting but not well introduced or discussed.
Response: We thank the Reviewer for the suggestion. We have described Antibody Resistance in Background section (Page 3, Lines 26-28). We agree that we have not explicitly discussed Antibody resistance term but our manuscript does discuss impact of the mutations on affecting binding to neutralizing antibodies.
Comment 7: Labels barely readable in many of the figures.
Response: The figures have been revised with larger fonts for clarity.